# Scoping review of food safety at transport stations in Africa

Busisiwe Purity Ncama,[1] Desmond Kuupiel ,[1,2] Sinegugu E Duma,[1] Gugu Mchunu,[1] Phindile Guga,[1] Rob Slotow[3]

[1]School of Nursing and Public Health, College of Health Sciences, University of KwaZulu-Natal, Durban, South Africa
[2]Research for Sustainable Development Consult, Sunyani, Ghana
[3]School of Life Sciences, College of Agriculture, Engineering and Science, University of Kwazulu-Natal, Pietermaritzburg, South Africa

**Correspondence to**
Dr Desmond Kuupiel;
desmondkuupiel98@hotmail.com

## ABSTRACT

**Objective** The WHO has declared food safety as a public health concern. Transport hubs such as taxi ranks, bus stations and other transport exchange sites are major food trading/purchasing sites, particularly in Africa. Research evidence is needed to improve food safety policies and ensure consumption of safe food, owing to the increasing burden of foodborne diseases, particularly in the WHO Africa Region. We systematically mapped and described research evidence on food safety at transport stations in Africa.

**Design** A scoping review guided by the Arksey and O'Malley framework.

**Data sources** We searched for original research articles in PubMed, Web of Science, and EBSCOhost (Academic search complete, CINAHL with Full-text and Health Source), SCOPUS, and Google Scholar from their inception to 25 October 2020.

**Eligibility criteria for selecting studies** We included studies that focused on food safety, involved transport stations, involved African countries and were published in English.

**Data extraction and synthesis** Data extraction was performed by two reviewers using a piloted-tested form. Thematic analysis was used to organise the data into themes and subthemes, and a narrative summary of the findings is presented.

**Results** Of the total 23 852 articles obtained from the database searches, 16 studies published in 6 countries met the inclusion criteria. These 16 studies were published between 1997 and 2019, with the most (5) in 2014. Of the 16 studies, 43.8% (7) were conducted in South Africa, 3 studies in Ghana, 2 in Ethiopia and 1 study each in Nigeria, Kenya, Lesotho and Zambia. Most (44.4%) of the included studies focused on microbial safety of food; few studies (22.2%) focused on hygienic practices, and one study investigated the perspective of consumers or buyers. Microbes detected in the foods samples were *Salmonella* spp, *Escherichia coli*, *Shigella* spp, *Bacillus* sp, *Staphylococcus aureus*, which resulted mainly from poor hygiene practices.

**Conclusions** There is limited research that focused on food safety at transport stations in Africa, especially on aspects such as hygiene practices, food storage and occupational health and food safety. Therefore, we recommend more research in these areas, using various primary study designs, to inform and improve food safety policies and practices for transport stations in African countries alongside improving access to clean water/handwashing facilities, and undertaking structural changes to facilitate behaviours and monitoring for unintended

## Strengths and limitations of this study

► To the best of our knowledge, this is the first scoping review to systematically explore literature and describe research evidence on food safety at transport stations as well as identify gaps for future research in Africa.
► This scoping review's evidence sources were searched using a systematic approach, and duplicate screening.
► This review is limited to Africa as well as English language publications.

consequences such as livelihoods of vulnerable populations.

## BACKGROUND

The WHO estimates that more than 600 million people fall sick (almost 1 in 10 people) with foodborne diseases annually, of which nearly 420 000 people die, and about 33 million years of healthy lives are lost every year worldwide.[1 2] The burden of foodborne diseases is estimated to be highest in the WHO African and South-East Asia Regions, mainly occurring among vulnerable populations such as infants, young children, pregnant women, older people, poor people and individuals with underlying illnesses.[3] Food contamination mostly results throughout the food supply chain (from the procedures used in processing the foods, inadequate storage temperatures, unhygienic practices by food handlers, poor sanitation at cooking places/vending areas, poor waste management and inadequate treatment of leftovers).[4]

Unsafe food has negative implications on health systems, and affects the development and national economies of countries, as well as trade.[3] Therefore, eating unsafe foods poses a significant public health threat. To avert the consequences of unsafe food on health systems, and to sustain national economies, development, trade and tourism,[5 6] the WHO in 2006 declared food safety as a global public health concern.[7 8] 'Food safety refers

to routines in the preparation, handling and storage of food meant to prevent foodborne illness and injury'.[5] To reduce the incidence of food-related diseases, particularly in high burden regions, the observations of food safety measures/precautions at all levels of the food processing chain, including the places where food is prepared and sold, are critical.[9 10]

Like other WHO Regions, especially in low-income and middle-income countries, food trading in the Africa Region takes place at several formal and informal places, such as in the markets, restaurants, streets, open spaces in academic institutions, and transport stations (taxi ranks, bus stations, lorry parks), and other transport exchange sites. Food vending at public spaces serves as a source of livelihood,[6 10 11] and more than two billion people eat food sold at various vending locations. including transportations stations on daily basis globally.[12 13] To this end, evidence is essential to inform in-country policies/guidelines, and further research, to ensure that food prepared and sold at transport stations promotes livelihoods, nutrition, food safety, and environmentally sustainable practices. This scoping review systematically mapped literature focused on food safety at transport stations in Africa, to summarise evidence and identify gaps.

## METHODS
### Scope of the review
The Arksey & O'Malley framework (research question identification; identifying relevant studies; selection of study; data charting, collating, and summarising and reporting the findings[14 15] was employed to scope and synthesise literature to answer the question—what evidence exists on food safety at transport stations in Africa? This review's study protocol was developed a priori.[16] This study included published peer-reviewed articles that reported findings from any African country/countries, focused on food safety, and involved transport stations. However, this study was limited to English publications (due to lack of expertise in other international languages), and primary study designs A detailed description of this scoping review study eligibility criteria is captured in the published protocol.[16] We followed the Preferred Reporting Items for Systematic Reviews and Meta-analyses (PRISMA) extension for Scoping Reviews checklist to report this study.[17]

### Identify relevant studies
We searched for primary research articles relating to food safety at transport stations in PubMed, Web of Science and EBSCOhost (Academic search complete, Cumulated Index to Nursing and Allied Health Literature (CINAHL) with Full-text, and Health Source), SCOPUS, and Google Scholar from their inception to 25 October 2020. To enable the capturing of all relevant articles, a comprehensive search strategy (developed in consultation with an expert librarian) consisting of keywords, Boolean terms (AND/OR), and Medical Subject Heading

terms, was used for the electronic database search (online supplemental file 1). Syntax was modified appropriately where needed. Filters such as date and study design were not applied during the literature search in the databases. DK and PG independently conducted the database search and title screening, and imported all potentially eligible articles onto an EndNote Library. The reference lists of all included articles were also screened for potentially relevant articles using the same approach.

### Selection of articles
Prior to the abstract screening, the 'find duplicates' function in EndNote was used to find all duplicate articles, and they were removed from the library. A screening form was developed in Google forms, using this study's eligibility criteria, for the abstract and full text screening phases. Two reviewers (coauthors) independently screened the abstracts as well as the full text articles. Discrepancies that arose during the abstract stage were resolved by discussion among the review team until a consensus was reached. At the full text screening phase, discrepancies were resolved by a third reviewer. All the additional articles identified from the reference list of the included articles equally underwent full text assessment. The PRISMA flow diagram was employed to account for all the articles involved.[18]

### Charting the data
A data extraction form was designed consisting of the following: author(s) and publication details, country of study, study design, study setting, study population, sample size, sex, study findings and recommendations. To ensure consistency and reliability, two reviewers piloted the data extraction sheet using a random sample of three included studies. The pilot testing of the form also enabled the review team to discuss discrepancies, and to revise the data extraction form prior to its final usage. Subsequently, 2 reviewers conducted the data extraction for the remaining 15 included studies using both inductive and deductive approaches. The review team resolved all discrepancies at this stage through discussion.

### Collating, summarising and reporting the results
This study subsequently employed thematic analysis, and collated all the emerging themes and subthemes relating to food safety. A summary of the findings from the included studies is presented narratively.

### Patient and public involvement
No patient involved.

## RESULTS
Of the 23 852 articles obtained from the database searches (see figure 1 flow diagram), 146 articles met the eligibility criteria at the title screening stage. Using EndNote "Find Duplicates" function, 30 duplicates were found and removed before abstract screening was

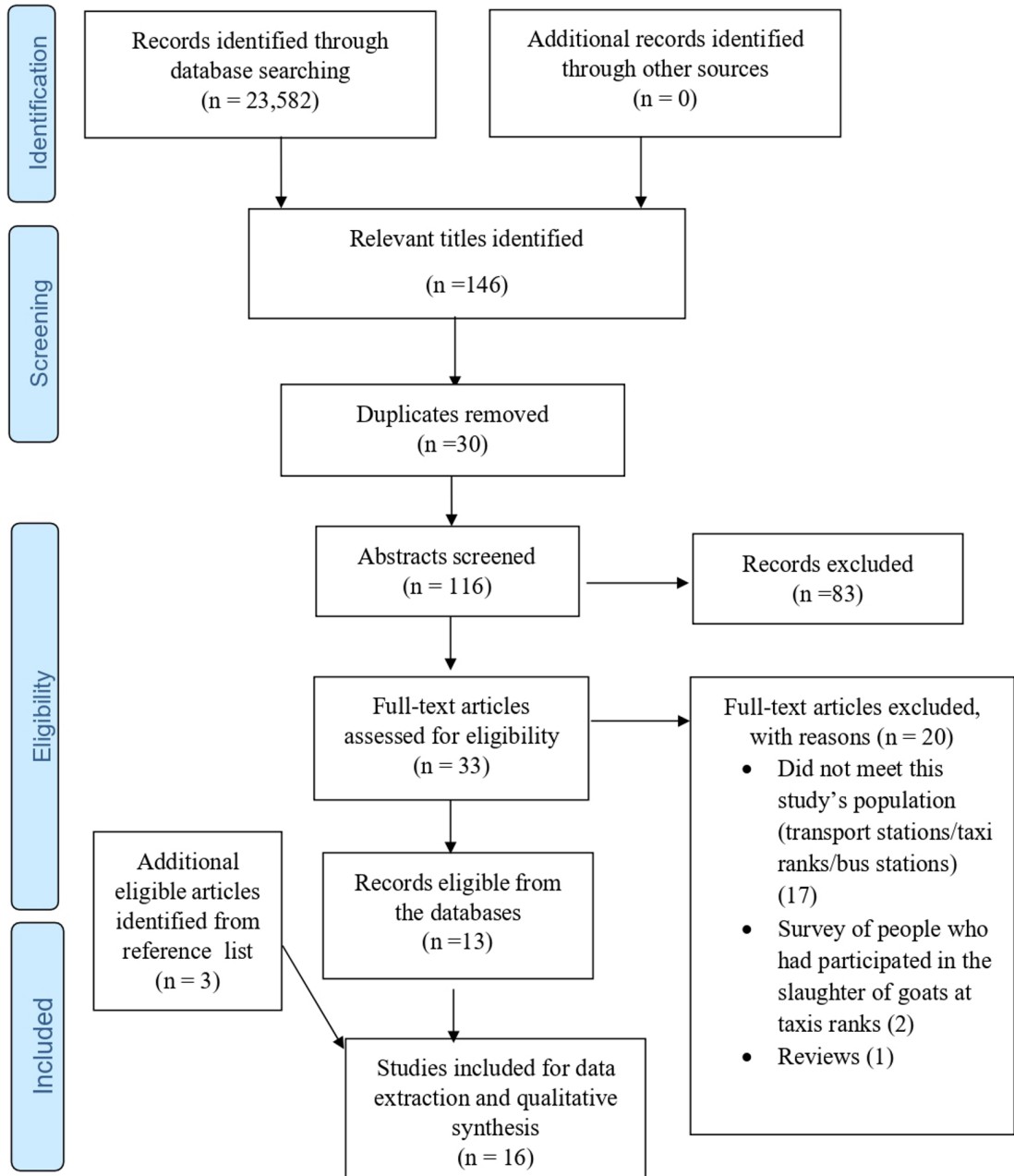

**Figure 1** Preferred Reporting Items for Systematic Reviews and Meta-Analyses 2009 flow diagram.

conducted. Subsequently, 83 articles were removed at the abstract screening, and 20 at full text (17 of these did not include transport stations/taxi ranks/ bus stations, but did involve sale from market centres, public places, chop bars, mini restaurants, major streets and sidewalks, and were excluded). Finally, 13 studies were included, and, from a manual search of their reference lists, a further 3 articles were added, giving a total of 16 articles for further analysis.

### Characteristics of the included studies

Table 1A,B presents a summary of the characteristics of the included studies. Of the 16 included studies, about 44 (43.8%) were conducted in South Africa,[19–25] 3 (18.8%) in Ghana,[4 26 27] 2 (12.5%) in Ethiopia[28 29]

and 1 (6.2%) each in Nigeria,[30] Kenya,[31] Lesotho[32] and Zambia.[33] Most of the studies were published in the last 6 years; however, no published study was found in 2015 and 2020 (figure 2). Fifteen (93.8%) of the included studies were cross-sectional studies, and one (6.2%) was a mixed-method study. Of the 16 included studies, 50.0% reported on microbial safety of food[4 19 23 27–29 33] and 25.0% reported hygiene practices of food handlers/ vendors.[6 21 30 31] One included study each reported on the following: occupational health and food safety risk[24]; knowledge of hygiene practice[26]; hygiene practices of food handlers/vendors and microbial safety[25]; and knowledge of food safety measures and hygiene practice by food handlers/vendors.[32]

**Table 1** Characteristic of the included sources of evidence

| Author, year | Country | City/town | Study design | Study setting | Study population | Sample size | Sex of vendors | Outcome reported |
|---|---|---|---|---|---|---|---|---|
| (A) | | | | | | | | |
| Oguttu et al, 2014[19] | South Africa | Tshwane Metropole, Gauteng Province | Mixed-methods study | Taxi rank | Vendors selling Ready-to-eat chicken | 100 samples of Ready-to-eat chicken | Females | Microbial safety of food |
| Mafune et al, 2016[20] | South Africa | Thohoyandou, Limpopo Province | Cross-sectional study | Taxi rank, bus station, shopping mall, and street stalls | Food samples from street vendors | 28 samples | Not specified | Microbial safety of food |
| Kibret and Tadesse, 2013[28] | Ethiopia | Bahir Dar Town | Cross-sectional study | Main roads sites, bus station, groceries, taxi ranks | Ready-to-eat white lupin sample from vendors | 40 samples (200 grams of white lupin) | Not specified | Microbial safety of food |
| Abakari et al, 2018[27] | Ghana | Tamale, Northern Region | Cross-sectional study | Taxi rank, bus stops, transport yard, and timber Market | Ready-to-eat salad samples from food vendors | 30 salad samples | Not specified | Microbial safety of food |
| Aluko et al, 2014[30] | Nigeria | Ile Ife, southwestern Nigeria | Cross-sectional study | Car parks | Food vendors | 160 (117 stationery and 43 mobile vendors) | Males and females | Hygiene practices of food handlers/vendors |
| Odundo et al, 2018[31] | Kenya | Not specified | Cross-sectional study | Major bus stops, markets, shopping areas, construction sites, and commercial areas | Food vendors | 130 | Males and females | Hygiene practices of food handlers/vendors |
| Kok and Balkaran, 2014[21] | South Africa | Durban, KwaZulu-Natal Province | Cross-sectional study | Transport exchange site | Food vendors | 29 | Not specified | Hygiene practices of food handlers/vendors |
| Letuka et al, 2019[32] | Lesotho | Maseru | Cross-sectional study | Taxi ranks | Food vendors | 141 (48 food handlers and 93 consumers) | Male and female | Knowledge of food safety measures and hygiene practice by food handlers/vendors |
| Eromo et al, 2016[29] | Ethiopia | Hawassa City | Cross-sectional study | Bus station | Food samples from street food vendors | 72 samples from six food items | Not specified | Microbial safety of food |
| (B) | | | | | | | | |
| McArthur-Floyd et al, 2016[26] | Ghana | Madina (Accra), Greater Accra Region | Cross-sectional study | Taxi rank, and transport exchange sites | Food vendors | 200 | Males and females | Knowledge of hygiene practice |
| Hill et al, 2019[6] | South Africa | Cape Town | Cross-sectional study | Train, bus stations, and taxi ranks, community centres, market | Food vendors | 831 | Males and females | Hygiene practices of food handlers/vendors |
| Mazizi et al, 2017[23] | South Africa | Alice (Nkonkobe) and King William's Town (Buffalo City), Eastern Cape province | Cross-sectional study | Taxi rank and bus stations | Street food vendors | 136 food samples-cooked and raw. | Not specified | Microbial safety of food |
| Qekwana et al, 2017[24] | South Africa | Tshwane Metropolitan Municipality, Gauteng Province | Cross-sectional study | Taxi ranks and Informal markets | Traditional goat slaughter | 105 people | Males and females | Occupational health and food safety risk |
| Flego and Sakyi, 2012[4] | Ghana | Kumasi, Ashanti Region | Cross-sectional study | Bus terminals | Food samples from vendors | 60 food samples | Not specified | Microbial safety of food |

Continued

**Table 1** Continued

| Author, year | Country | City/town | Study design | Study setting | Study population | Sample size | Sex of vendors | Outcome reported |
|---|---|---|---|---|---|---|---|---|
| Tshipamba et al, 2018[25] | South Africa | Johannesburg | Cross-sectional study | Taxi ranks and streets | Meat samples from vendors | 115 meat samples | Not specified | Hygiene practices of food handlers/vendors, and microbial safety of food |
| Jermini et al, 1997[33] | Zambia | Not specified | Cross-sectional study | Bus park/station and large market | Samples of raw, processed, and cooked Foods from street food vendors | Not specified | Not specified | Microbial safety of food |

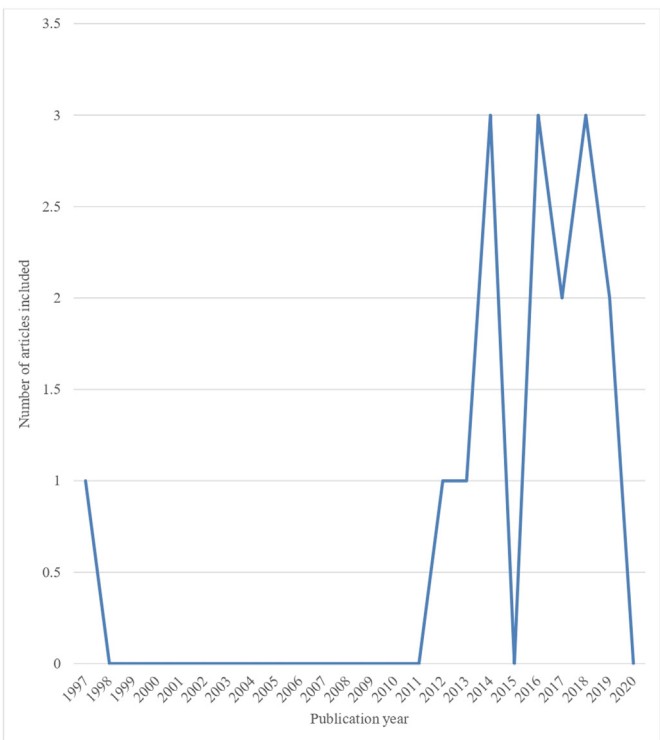

**Figure 2** Trend of published studies relating to food safety at transport station in Africa.

### Findings from the included studies

#### Microbial safety of food

Of the nine included studies that reported findings on microbial safety of food, 44.4% were conducted in South Africa,[19 20 23 25] 22.2% each in Ghana[4 27] and Ethiopia[28 29] and the last 11.1% in Zambia.[33] Seven of the eight studies reported unacceptable levels of microbes in the food.[4 19 20 23 27–29 33] Table 2A–C presents a summary of the key findings as well as the sample type, analytic approach and the microbes reported.

#### Hygiene practices of food handlers/vendors

#### Food preparation

Of the 16 included studies, 8 reported research findings relating to food preparation. Fifty per cent of these eight studies were from South Africa[6 21 23 25] and the remainder were from Ghana,[26] Nigeria,[30] Kenya[31] and Lesotho.[32] The studies in South Africa focused on the following: hygiene practices and implications for consumers[21]; food and nutrition knowledge as well as practices related to food preparation,[7] the effect of hygiene practices and attitudes of meat vendors,[25] and sources of food contamination.[23] The study from Ghana investigated how fast food operators washed their hands,[26] while the studies from Nigeria, Kenya and Lesotho evaluated food safety and sanitary practices[30]; food vendors and hygiene practices; and food safety knowledge, attitudes and practices[31] of food vendors and consumers' perceptions.[32] A summary of the key findings from these studies is presented below (table 3).

**Table 2** Microbial safety of food at transport stations

| Study | Type of sample | Analytical approach | Microbes reported | Key results | Key conclusion |
|---|---|---|---|---|---|
| (A) | | | | | |
| Oguttu et al[19] | Ready-to-eat (RTE) chicken | 3M Petrifilm plates | Staphylococcus aureus | High prevalence of S. aureus in the sample was (44%; 90% CI 36.1% to 52.2%), with mean S. aureus counts of 103.6 (90% CI 103.3 to 103.9). The likelihood of food poisoning with S. aureus from RTE chicken was estimated to be 1.3% (90% CI 0% to 2.7%) | To reduce the levels of concentration of S. aureus on the RTE chicken and promote the sale of safer and affordable RTE chicken for the large urban poor population in South Africa, training of RTE chicken vendors on hygiene is still needed. |
| Mafune et al[20] | Unfermented porridge, boiled cabbage and carrots, boiled peanuts, salad, potato chips, traditional mageu, and stewed beef and grilled chicken | Standard microbiological method | S. aureus | S. aureus was <2.4771 log10 cfu/g in all samples and places. Except for fried potato chips, microbial contamination was observed in the remaining food samples using the total plate count method. | Most of the vended foods investigated met the microbiological standard of RTE foods |
| Mazizi et al[23] | Cooked and raw beef, pork, and mutton samples, surface contact plates, and water samples | Biochemical tests according to international standards methods | S. aureus, Escherichia coli, and Salmonella spp. | Mean score of raw beef, mutton, and pork were aerobic plate counts (4.8, 3.7 and 2.8 Log (cfu/g)), S. aureus (3.3, 3.7 and 2.8 Log cfu/g), and E. coli (1.0, 0.6 and 0.3 Log cfu/g) respectively. | The levels of contamination in cooked meat were lower when compared with the standards set by Commission Regulation for determining the microbiological quality of RTE foods. |
| Tshipamba et al[25] | RTE meat | Standard biochemical and molecular methods | Bacillus thuringiensis, Bacillus spp., Bacillus subtilis, Bacillus cereus, Citrobacter spp., Enterococcus faecium, Enterococcus faecalis, Kurthia spp., Lysinibacillus spp., Macrococcus caseolyticus, Planomicrobium glaciei, Planococcus antarcticus, S. aureus, S. equorum, and S. vitulinus | Overall mean total bacteria in the samples ranged from 4.3 to 6.03 cfu/mL×102 and coliform counts ranged from 1.60 to 1.95×102 cfu/mL Of the 15 microbes identified, S. aureus occurred in all the meat types and the percentage of occurrence was chicken meat (14%), beef head (43%), beef intestine (50%), and wors (sausage) (20%) | Consumers RTE meat are at risk of food borne diseases due to poor hygiene practices of the vendors. |
| (B) | | | | | |
| Kibret and Tadesse[28] | White lupin | Standard bacteriological techniques, and Kirby-Bauer disk diffusion method for antimicrobial susceptibility test | E. coli, Salmonella spp, and Shigella spp. | Prevalence of bacteria total coliform counts were 954.2±385 at the surface and 756.2±447.3 at the core of white lupin. Pathogens isolated were as follows E. coli 29 (72.5%), Salmonella spp. 23 (57.5%) and Shigella spp. 8 (20%). Overall multiple antimicrobial resistances rate was 75% | Contamination of white lupin and a potential health risk to consumers revealed, and the bacteria isolated showed high rates of multiple drug resistance. |

Continued

**Table 2** Continued

| Study | Type of sample | Analytical approach | Microbes reported | Key results | Key conclusion |
|---|---|---|---|---|---|
| Eromo et al[29] | Local bread ('ambasha' and 'kita'), raw fish, chilli ('awaze'), avocado, and cooked potato | Standard microbiological techniques | E. coli, Salmonella spp., and S. aureus | The microbiological quality in nearly 31% of RTE food samples was beyond the acceptable limits. Total colony counts detected ranged from 1.7×105 to 6.7×106 cfu/g. E. coli (29.6%), Salmonella spp. (12.7%, and S. aureus (9.9%) were the most frequent isolates. All isolates were 100% sensitive to ciprofloxacin, but 89% of Salmonella spp. was resistant to chloramphenicol, 14.3% of S. aureus was resistant to vancomycin | Considerable rate of contamination in the foods confirmed. The identified foodborne bacteria and antibiotic resistance isolates could pose a public health problem in the study location. |
| Abakari et al[27] | Precut vegetable salads | Standard microbiological methods | E. coli, Bacillus cereus, Salmonella spp. and Shigella spp. | E. coli levels ranged from 0 to 7.56 log10 cfu/g; Bacillus cereus levels ranged from 0 to 7.44 log10 cfu/g; Salmonella spp. ranged from 0 to 4.54 log10 cfu/g, and Shigella spp. ranged from 5.54 log10 cfu/g were detected in 96.7%, 93.3%, 73.3%, and 76.7% of the salads samples, respectively. | Salads were revealed to be unwholesome for human consumption and could be deleterious to the health of consumers. |
| (C) | | | | | |
| Flego and Sakyl[4] | RTE foods (ice-kenkey,[15] cocoa drink,[15] fufu,[5] RTE red pepper for kenkey),[5] salad,[10] and macaroni[10]) | Standard microbiological methods | Staphylococci, Bacillus spp., Klebsiella pneumoniae, Aeromonas pneumophila, E. cloacae, S. aureus, E. coli, and P. aeruginosa | RTE foods were found to be contaminated with enteric bacteria and other potential food poisoning organisms with bacterial counts higher than the acceptable levels (<5.0 log10 CFU/mL). Coagulate negative staphylococci (23.7%), Bacillus species (21.5%), K. pneumoniae (18.0%), Aeromonas pneumophila (17.7%), E. cloacae (6.7%), S. aureus (3.7%), E. coli (2.2%) and P. aeruginosa (2.2%) were the main isolates detected. | Most RTE foods were contaminated with enteric bacteria and other potential food poisoning organisms with bacterial counts higher than the acceptable levels. |
| Jermini et al[33] | Raw foods (ground meat, chicken, and chicken intestine); and processed foods (dried 'minnows' and 'kapenta') | | Salmonellae Spp., S. aureus, Clostridium peifringens | Raw foods such as ground meat, chicken, chicken intestine; and processed foods such as dried minnows and kapenta were contaminated by salmonellae or contained high populations of S. aureus in pasteurised milk. High populations (>105) of S. aureus were detected from a sample of leftover chicken, more than 107 were detected in leftover rice, and 10 million C. peifringens per gram were detected in leftover beef stew sample | Time–temperature exposures during reheating had variable effects in terms of killing the microorganisms that germinated from surviving spores or that reached the foods after cooking. |

**Table 3** Key reported findings on food preparation

| Study | Key findings reported |
|---|---|
| Kok and Balkaran[21] | ▶ Water being used for washing utensils was left unchanged,<br>▶ Piles of dirty pots and dishes was left near the serving areas and RTE foods, and garbage left uncovered with many flies at the site,<br>▶ RTE food was left uncovered,<br>▶ Most of the food handlers were not wearing gloves, hairnets, or aprons |
| Hill et al[22] | ▶ 85.5% of the vending stalls lacked soap or surface sanitizer,<br>▶ 71% lacked basin for washing,<br>▶ 75% did not have drying cloth,<br>▶ 76.6% of vendors handled food and money concurrently,<br>▶ About 57% left the food uncovered.<br>▶ 39% of the vendors were using their hands to pick up food items, with only 6% wearing gloves, and<br>▶ 29% of vendors had a wet clean sponge/cloth obtainable at the site |
| Mazizi et al[23] | ▶ Major sources of food contamination identified were poor hygiene practices of the food vendor, holding area, and the utensils |
| Tshipamba et al[25] | ▶ Approximately 90% of RTE meat vendors at the taxi rank exposed their meats to dust and flies,<br>▶ 94% of them handled money while serving food, and<br>▶ Stagnant water found in about 22% of the vending locations at the taxi-rank |
| McArthur-Floyd et al[26] | ▶ 64% of food vendors washed their hands from elbow to finger and the remainder (36%) washed from their wrist to finger (the WHO recommends handwashing from elbow to fingers), and<br>▶ 62% of the vendors test their meal in the palm while 38% of them test it with a spoon (the best way to test a meal) |
| Aluko et al[30] | ▶ Approximately 17% of food vendors washed their hands always after using the toilet,<br>▶ 63% of them rarely kept their fingernails short, and<br>▶ Nearly 4% of them always kept their leftover cooked food in a refrigerator, despites having unstable power supply |
| Odundo et al[31] | ▶ Food vendors had poor hygiene practices however, men were observed to have better hygienic practices than women (p<0.05),<br>▶ Hygiene practice of the vendors was found to be significantly associated with training (those trained observe hygiene), and<br>▶ Wearing of jewellery, long and unclean nails, and lack of protective clothing were observed. |
| Letuka et al[32] | ▶ Observed that the food handlers operated under unhygienic environment |

RTE, ready-to-eat.

## Knowledge of hygiene practices/food safety precautions

In Ghana, McArthur-Floyd et al study[26] revealed that the majority (94%) of fast food operators knew food safety precautions.[26] Letuka et al study[32] in Lesotho indicated that 95% of food vendors did not know washing utensils with detergents helps reduce contamination.[32] The mean knowledge (49%±11) of the food vendors included in the study was considered poor.[32] About 6% of the consumers that participated in the study chose not to buy food sold at taxi ranks due to food safety issues and hygiene.[32]

## Occupational health and food safety risk

In South Africa, Qekwana et al[24] evaluated the occupational health and food safety risks associated with the traditional slaughter of goats, and the consumption of such meat.[24] Approximately 63% of the practitioners were not wearing protective clothing during slaughter, and about 78% of practitioners did not know their health status.[24] Almost 83% of the practitioners hung up their carcass to facilitate bleeding, flaying and evisceration.[24] The study further observed that none of the practitioners practiced

meat inspection. In Nigeria, Aluko et al[30] study revealed that approximately 62% of the vendors had no formal training, and their medical status was also unknown.[30]

## DISCUSSION

This scoping review mapped evidence on food safety at transport stations in Africa, and revealed a very low number of papers that are published in this area, given many African employees in both formal and informal sectors commute through these transport hubs.[12 13] An average of one paper per year relating to food safety at transport hubs in Africa as revealed by this review is simply not enough. Nonetheless, the few papers depict an imbalance of research, with most focused on microbial safety,[4 19 20 23 27–29 33] and few on socioeconomic aspects such as hygiene practices,[6 21 30 31] and occupational health and food safety risk.[24] Moreover, this review revealed no study evaluated the storage of food or how the food is transported to the vending site.

As evidence by this review, most of the food sold at transport hubs does not meet the minimum standards and is not safe for consumption due to the presence of several microbes.[4 19 23 25 27 29 33 34] There are several reasons for this such as poor practices relating to hygiene, storage, preparation, cooking, cleaning and serving.[4 19 20 23 27–29 33] However, these findings are similar to previous review findings involving markets,[35] homes and restaurants.[36] A recent publication by Gizaw[35] indicated that several studies reported microbial contamination of foods sold in the market, with bacteria and fungi similar to those identified in our review.[35] Also, a review by the WHO reported that the main factors contributing to foodborne disease outbreaks in homes or restaurants were poor temperature control in preparing, cooking, and storing food.[36] Although very few papers were found by this review, the evidence is compelling that there should be policy interventions to address issues relating poor hygiene practices, including food storage, preparation, cooking, cleaning and serving by food handlers at transport hubs, not only in South Africa but across Africa.

Similar to a previous scoping review[10] most of the included papers were published within the last 6 years but, no published study was found in 2015 and 2020. While the reason for the lack of published papers in 2015 might be difficult to determine, the COVID-19 pandemic which resulted in 'covidisation' of research might be the reason for the lack of publication in this field of research in 2020. Although we cannot conclude that no primary research has been conducted in these countries focusing on the safety of food sold at transport stations, it suggests a research/publication gap. Food safety research is, perhaps, more relevant now than ever in Africa, since the burden of foodborne diseases is rising annually, resulting in the declaration of food safety as a public health concern by the WHO.[7 8] Aside from this, most commuters tend to buy ready-to-eat (RTE) food from street food vendors, including those at transport hubs[37 38]; hence, the sale of food at transport stations is rising,[38 39] particularly in Africa[6] partly due to an increase in demand for RTE, and the employment opportunities it offers to many individuals who otherwise would not have had any source of income.[10 40] Even more worrying is the fact that most of the articles included that focused on microbial safety, reported high levels of food contamination with several microorganisms, especially *Salmonella* spp and *E. coli.*[4 19 23 25 27 33 34] Therefore, more research is needed across African countries to prevent potential negative consequences.

Our study findings have implications for practice and research. For instance, the likelihood of food poisoning with microbes such as *Salmonella* spp, *E. coli., Shigella* spp, *Bacillus* spp, *S. aureus* and several others, revealed by most of the included studies that focused on microbial contamination of food, is alarming. This, if not checked, could further worsen the already high burden of foodborne diseases in a continent that has several of its countries already experiencing many health systems and economic challenges. Aside from this, the majority of individuals who commute through transport hubs, possible will purchase a meal from a transport hub/exchanges site, which may be the only meal[12 13] of the day and yet the food safety standards are poor.[4 19 20 23 27–29 33] Thus, if not checked, the excess cases of foodborne diseases from any outbreak will further impact negatively on the already challenged public health systems in Africa. Also, poor people who are exposed to these unsafe foods get an infection, may have to pay more for healthcare, which can further exacerbate their poverty situation. Moreover, people who are already living in extreme poverty who get exposed to foodborne disease may not even make it to the hospital for care and can end up dying at home.[41]

Good hygiene and sanitation practices, such as adequate hand washing, adequate washing and storage of pots and dishes, good waste management, observation of food preparation standards and serving etiquette, among others, have the potential to reduce the risk of food contamination from both biological and non-biological hazards, yet this study reveals fewer studies that focused on hygienic practices. We, therefore, recommend more research to further inform contextualised policy decisions aimed at improving hygiene and sanitation practices by food vendors at transport stations. Also, very relevant to ensuring food safety is the occupational health practices of the vendors. Regular food handling tests and food inspections, conducted by the appropriate local authorities, should be mandatory in all African countries. Food handler tests should seek to ensure that food vendors are fit healthwise to prepare and serve food meant for public consumption. However, our review found limited studies that evaluated occupational health and food safety. Considering that evidence from South Africa and Nigeria suggests about 78% and 62% of food vendors do not know their health status[30 42] and the increasing number of informal food sellers at various transport exchange sites, future studies are recommended to focus on occupational health and food safety in Africa. The means and manner of storing food, especially leftover RTE food, can either increase or reduce the risk of food contamination, but, again, this scoping review found no study that focused on food storage practices of the vendors at transport stations. Also essential, and yet we did not find any study focusing on it, is the quality of food (nutritious aspects) of the meals sold at transport stations. Eating a well nourishing diet or balanced meals is critical to ensure good health[43–45]; hence, we encourage future primary studies to include the nutritious aspects. Such studies may help streamline guidelines or inform policies to improve the quality of the food sold at transport exchange sites or taxi ranks. Moreover, this review found that the majority (17 out of 18) of the respondents in the included studies were the vendors (mostly women) or food samples taken from the vendors. The perspectives of consumers (buyers) or commuters regarding food safety at transport stations are

also very relevant, and we recommend future research to involve them. A comparative study to investigate food safety practices among males and females food vendors at transport stations might be relevant since many males are now getting involved in the business.[6 46 47]

To the best of our knowledge, this study is the first scoping review that systematically mapped literature relating to food safety at transport stations in Africa. A major strength of our study method is that it permits the inclusion of multiple study designs. Also, the choice of this study method permitted us to highlight literature gaps, and made recommendations for future research. Aside from this, we conducted a thorough search in six databases using a comprehensive search strategy which enabled us to capture the most relevant articles to answer the review question. Moreover, two independent reviewers were used to select the studies and perform data extraction processes which helped to prevent selection bias and ensured the reliability and trustworthiness of this study results. Despite this, our scoping review has many limitations. This study included only original study peer reviewed papers, which resulted in the exclusion of one review paper[10] and one Masters' dissertation.[48] We did not also consult the websites of WHO and the Food and Agriculture Organisation websites for possible relevant studies. Furthermore, this study cannot be generalised since the search was limited to African countries only. Although date limitation was removed, we limited the publication language to English only, which perhaps eliminated relevant articles published in other languages. Despite these limitations, this study has provided essential evidence relating to food safety at transport stations and has shown literature gaps to guide future research.

## CONCLUSION

Based on this scoping review's eligibility criteria, our study results suggest there is limited research focusing on food safety at transport stations in Africa. Most of the existing published studies are focused on microbial safety of food, and very few/none on other aspects such as hygiene practices, food storage, occupational health and food safety, and nutrition. Hence, we recommend more primary research involving community members and policymakers in these areas going forward alongside improving access to clean water/handwashing facilities, and undertaking structural changes to facilitate behaviours and monitoring for unintended consequences such as livelihoods of vulnerable populations.

**Contributors** BPN, DK, SED, SM and RS conceptualised and designed the study. DK developed and designed the database search strategy and conducted the search. PG contributed to the screening of the studies and data extraction. DK wrote the draft manuscript and BPN, SED, GM and RS critically review it and made revisions. All the authors approved the final version of the manuscript. BPN is the author responsible for the overall content of this study.

**Funding** Funding for this work was provided by the Sustainable and Healthy Food Systems (SHEFS) Programme, supported through the Wellcome Trust's Our Planet,

Our Health Programme [grant number: 205200/Z/16/Z]. The funder played no role in the literature search and writing of the manuscript.

**Competing interests** None declared.

**Patient consent for publication** Not applicable.

**Provenance and peer review** Not commissioned; externally peer reviewed.

**Data availability statement** Data are available upon reasonable request. Data sharing not applicable as no datasets generated and/or analysed for this study. All data relevant to the study are included in the article or uploaded as supplementary information.

**ORCID iD**
Desmond Kuupiel http://orcid.org/0000-0001-7780-1955

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
