## [Reviewer comments · BMJ Open]

ARTICLE DETAILS

TITLE (PROVISIONAL)	A Scoping Review of Food Safety at Transport stations in Africa
AUTHORS	Ncama, Busisiwe; Kuupiel, Desmond; Duma, Sinegugu; Mchunu, Gugu; Guga, Phindile; Slotow, Rob

VERSION 1 – REVIEW

REVIEWER	Hoffmann, Vivian International Food Research & Policy Institute (IFPRI)
REVIEW RETURNED	13-Jun-2021

GENERAL COMMENTS	This is an interesting paper on an important topic which, as the authors indicate, deserves more research attention. Main comments: 1. I am curious about why the authors chose to focus exclusively on transport hubs. What makes these locations for the sale of prepared foods different from other sites such as markets and sidewalks? While I appreciate the need to focus the review, only 18 articles met the inclusion criteria. The number of studies reporting on microbial contamination (9) or food hygiene practices (8) are fewer yet. Unless there is a strong reason to focus on transport hubs specifically, it seems to me that including the other 16 studies identified by the authors on the safety of street-vended food that were omitted due to a non-transport hub location, would enable a richer and more informative review.2. The two studies by Qekwana et al., "Assessment of the occupational health and food safety risks associated with the traditional slaughter and consumption of goats in Gauteng, South Africa", and "Designing a risk communication strategy for health hazards posed by traditional slaughter of goats in Tshwane, South Africa." do not describe food safety at transport stations. Rather, they rely on a survey of a convenience sample of people who had participated in the slaughter of goats, who were surveyed at taxis ranks about this activity. They should not be included in this review.3. The authors claim in the abstract and discussion section that "most of the food sold at transport hubs do not meet the minimum standards, and is not safe for consumption due to the presence of several microbes". This seems to be an overstatement, as the key results on microbial contamination shown in Table 3 indicate that for most of the studies that reported the proportion of samples in which microbial contamination was or above the relevant level of concern, this was below 50%.
--

	4. The authors claim it is “worrying” that none of the studies reviewed “looked at the nutritious aspects of meals sold, despite an established prevalence of poor nutrition and ill-health”. But studies not selected based on the topic of food safety, not nutrition – so it is unreasonable to expect they should also cover this topic. A separate search would potentially identify many more articles on the nutrition quality of foods sold at transport hubs. 5. I don't see any information on eligibility criteria in the section on "Selection of articles and edibility criteria". Also there is a typo in this section title. Other comments: 1. “Food safety consists of food preparation, handling, storage, and hygienic practices aimed to prevent food contamination by microbial, chemical, and physical hazards in the food production chain”: These seem like a list of food safety practices, rather than a definition of food safety itself. 2. "This scoping review systematically mapped literature focused on food safety at transport stations in Africa, to provide research evidence and gaps": Better words would be to “summarize” evidence and “identify” gaps? "Letuka et. al. study in Lesotho, indicated that 95% of food vendors had incorrect knowledge that washing utensils with detergent leave them free of contamination": this seems like an odd detail from that study to highlight; while not guaranteed to eliminate contamination, washing with detergent is a recommended practice.
--	--

REVIEWER	Mohammad, Zahra University of Houston, Hotel and Restaurant Management
REVIEW RETURNED	15-Jun-2021

GENERAL COMMENTS	The manuscript reviews food safety at transport stations in Africa. Generally speaking, the manuscript describes potential food safety risks related to the food purchased at transport station points, regulatory and knowledge gaps. While the information is very helpful for food safety educators, researchers, policymakers, the paper needs some revision prior to publishing, especially with regard to discussion. Authors should focus on reducing redundancy within the discussion section. In addition, thorough revision is needed for grammar and spelling errors. The following are some comments, but the manuscript loaded with many other grammatical issues, punctuation, and spelling errors. Line 32, Organization not organization Line 155, It is unclear how the authors contacted reviewers; are these reviewers co-authors or independent reviewers? Please clarify! Line 172, What is the eligibility criteria? Authors should include a brief description here about eligibility criteria to help readers understand the base of their search and targeted literature. line 289 S. aureus not S. aureu
---

VERSION 1 – AUTHOR RESPONSE

Responses to Reviewer: 1 (Dr. Vivian Hoffmann) comments

Comments to the Author:

This is an interesting paper on an important topic which, as the authors indicate, deserves more research attention.

Response

On behalf of the authors, I wish to extend our gratitude to you for making time out of your busy schedule to review this manuscript and providing us with useful comments to improve the paper. Please see below our point-by-point responses to your comments/suggestions for your consideration.

Main comments:

Comment 1

I am curious about why the authors chose to focus exclusively on transport hubs. What makes these locations for the sale of prepared foods different from other sites such as markets and sidewalks? While I appreciate the need to focus the review, only 18 articles met the inclusion criteria. The number of studies reporting on microbial contamination (9) or food hygiene practices (8) are fewer yet. Unless there is a strong reason to focus on transport hubs specifically, it seems to me that including the other 16 studies identified by the authors on the safety of street-vended food that were omitted due to a non-transport hub location, would enable a richer and more informative review.

Response

We sincerely appreciate your concern, but the funding obtained was meant to investigate food safety at transport hubs exclusively hence our inability to expand it to include markets or sideways per your suggestion.

Comment 2

The two studies by Qekwana et al., "Assessment of the occupational health and food safety risks associated with the traditional slaughter and consumption of goats in Gauteng, South Africa", and "Designing a risk communication strategy for health hazards posed by traditional slaughter of goats in Tshwane, South Africa." do not describe food safety at transport stations. Rather, they rely on a survey of a convenience sample of people who had participated in the slaughter of goats, who were surveyed at taxis ranks about this activity. They should not be included in this review.

Response

We have agreed to exclude them upon a second look and all the needed amendments done throughout the paper. Thank you for your comment.

Comment 3

The authors claim in the abstract and discussion section that "most of the food sold at transport hubs do not meet the minimum standards, and is not safe for consumption due to the presence of several microbes". This seems to be an overstatement, as the key results on microbial contamination shown in Table 3 indicate that for most of the studies that reported the proportion of samples in which microbial contamination was or above the relevant level of concern, this was below 50%.

Response

Thank you for your observation. Please we have revised to remove the statement.

Comment 4

The authors claim it is "worrying" that none of the studies reviewed "looked at the nutritious aspects of meals sold, despite an established prevalence of poor nutrition and ill-health". But studies not selected based on the topic of food safety, not nutrition – so it is unreasonable to expect they should also cover this topic. A separate search would potentially identify many more articles on the nutrition quality of foods sold at transport hubs.

Response

We have removed the statement.

Comment 5

I don't see any information on eligibility criteria in the section on "Selection of articles and edibility criteria". Also there is a typo in this section title.

Response

Thank you for drawing our attention to this error. Please we have modified the title to read "Selection of articles" (Page 5, LN 147). This study eligibility criterion is captured in the scope of the review section (Page 5, LN 125-129).

Other comments:

Comment 1

"Food safety consists of food preparation, handling, storage, and hygienic practices aimed to prevent food contamination by microbial, chemical, and physical hazards in the food production chain": These seem like a list of food safety practices, rather than a definition of food safety itself.

Response

Please we have revised it (Page 4, LN 102-103).

Comment 2

"This scoping review systematically mapped literature focused on food safety at transport stations in Africa, to provide research evidence and gaps": Better words would be to "summarize" evidence and "identify" gaps?

Response

Thank you for your suggestion. We have revised it accordingly (Page 4, LN 116-117).

Comment 3

"Letuka et. al. study in Lesotho, indicated that 95% of food vendors had incorrect knowledge that washing utensils with detergent leave them free of contamination": this seems like an odd detail from that study to highlight; while not guaranteed to eliminate contamination, washing with detergent is a recommended practice.

Response

Please we have revised the statement to clarity (Page 8, LN 221-223)

Responses to Reviewer: 2 (Dr. Zahra Mohammad) Comments

Comments to the Author:

Comment 1

The manuscript reviews food safety at transport stations in Africa. Generally speaking, the manuscript describes potential food safety risks related to the food purchased at transport station points, regulatory and knowledge gaps. While the information is very helpful for food safety educators, researchers, policymakers, the paper needs some revision prior to publishing, especially with regard to discussion. Authors should focus on reducing redundancy within the discussion section. In addition, thorough revision is needed for grammar and spelling errors.

The following are some comments, but the manuscript loaded with many other grammatical issues, punctuation, and spelling errors.

Response

On behalf of the authors, I wish to extend our gratitude to you for making time out of your busy schedule to review this manuscript and providing us with useful comments to improve the paper. We have made revisions in the discussion section in addition to thorough proofreading to correct all grammatical and spelling errors identified per your comment.

Comment 2

CLine 32, Organization not organization

Response

Please we have corrected it (Page 2, LN 32).

Comment 3

Line 155, It is unclear how the authors contacted reviewers; are these reviewers co-authors or independent reviewers? Please clarify!

Response

Please they co-authors, but the screening was done independently. See page 5, LN 150 for clarification.

Comment 4

Line 172, What is the eligibility criteria? Authors should include a brief description here about eligibility criteria to help readers understand the base of their search and targeted literature.

Response

This study's eligibility criterion is captured in the "scope of the review" section (Page 5, LN 125-129).

Comment 5

line 289 S. aureus not S. aureu

Response

Thank you. We have corrected it (Page 9, LN 279).